# Analysis of the Presence of the Virulence and Regulation Genes from *Staphylococcus aureus* (*S. aureus*) in Coagulase Negative Staphylococci and the Influence of the Staphylococcal Cross-Talk on Their Functions

**DOI:** 10.3390/ijerph20065155

**Published:** 2023-03-15

**Authors:** Magdalena Grazul, Ewa Balcerczak, Monika Sienkiewicz

**Affiliations:** 1Department of Pharmaceutical Microbiology and Microbiological Diagnostics, Medical University of Lodz, Muszynskiego 1 Street, 90-151 Lodz, Poland; 2Laboratory of Molecular Diagnostics and Pharmacogenomics, Department of Pharmaceutical Biochemistry and Molecular Diagnostics, Medical University of Lodz, Muszynskiego 1 Street, 90-151 Lodz, Poland

**Keywords:** staphylococcus, genes transfer, virulence, coagulase negative, CoNS virulence

## Abstract

Coagulase-negative staphylococci (CoNS) are increasingly becoming a public health issue worldwide due to their growing resistance to antibiotics and common involvement in complications related to invasive surgical procedures, and nosocomial and urinary tract infections. Their behavior either as a commensal or a pathogen is a result of strict regulation of colonization and virulence factors. Although functionality of virulence factors and processes involved in their regulation are quite well understood in *S. aureus*, little is known about them in CoNS species. Therefore, the aim of our studies was to check if clinical CoNS strains may contain virulence factors and genes involved in resistance to methicillin, that are homologous to *S. aureus.* Moreover, we checked the presence of elements responsible for regulation of genes that encode virulence factors typical for *S. aureus* in tested isolates. We also investigated whether the regulation factors produced by one CoNS isolate can affect virulence activity of other strains by co-incubation of tested isolates with supernatant from other isolates. Our studies confirmed the presence of virulence factor and regulatory genes attributed to *S. aureus* in CoNS isolates and indicated that one strain with an active *agr* gene is able to affect biofilm formation and δ-toxin activity of strains with inactive *agr* genes. The cognition of prevalence and regulation of virulence factors as well as antibiotic resistance of CoNS isolates is important for better control and treatment of CoNS infections.

## 1. Introduction

The ability of staphylococci to occupy numerous host niches, and exist either as a commensal or a pathogen has been attributed to a strict regulation of colonization and virulence factors, e.g., haemolysins, toxins, adhesins [1].

For many years, only *S. aureus* was considered as pathogenic among the staphylococci group, but lately it is postulated that coagulase-negative staphylococci (CoNS) may produce various virulence factors typically attributed to *S. aureus,* e.g., haemolysins, enterotoxins, and thus there may be ethological factors of infections. Therefore, CoNS species also need more scientific attention related to their pathological capacity. Although functionality of virulence factors and processes involved in their regulation are quite well understood in *S. aureus*, little is known about them in CoNS species [2].

Some *S. aureus* strains often contain in their genomes virulence genes that are not found in all strains and that may be carried on discrete genetic elements. Genome analyses revealed that there are different sets of genes encoding different virulence factors in *S. aureus,* as many of these genes are located in pathogenicity islands wherein genes are translocated by horizontal genome transfer (HGT) via phage transduction, conjugation or by direct uptake of naked DNA by genetic competence. The presence/absence of certain genes on the pathogenicity island of any strain may be a consequence of unique horizontal gene transfer events that occurred within specific *S. aureus* lineages; in this way, a non-virulent strain may become pathogenic. As all staphylococci, not only *S. aureus*, may contain similar genes or ability to acquire genes from related species by a horizontal gene transfer (HGT) process, thus, CoNS also become increasingly infective. Therefore treatment of infections caused by CoNS is becoming more and more difficult due to the growing rates of resistance to antibiotics and virulence [2]

Particularly, *S. aureus* but also coagulase-negative (CoNS) staphylococci are able to secrete numerous exotoxins (alpha, beta, gamma, and delta) that invade host cells. Most *S. aureus* strains are able to produce at least three (haemolysins HlgAB, HlgCB, and leukocidins LukAB/HG) of the six known leucocidins, and many highly virulent clinical strains able to affect humans produce five (HlgAB, HlgCB, LukAB/HG, Panton–Valentine leucocidin [PVL], and LukED) toxins. γ-haemolysins are divided into two types of bicomponent elements, namely, HlgAB and HlgCB. Both of them consist of units from class S: HlgA (class S component encodes gamma haemolysin-A-like protein), or HlgC (encodes gamma haemolysin C) respectively and HlgB unit (encodes gamma haemolysin-B) from class F. These proteins, after recognition of their cell targets, undergo conformational changes and form oligomeric complexes. This process causes trans-membrane pore formation, and therefore leads to cell death. HlgAB toxin is involved in blood stream infections, e.g., bacteremia and septic arthritis [3], and pathogenic processes of toxic shock syndrome (TSS) as it affects polymorph nuclear cells, monocytes, macrophages, and erythrocytes. HlgCB is responsible for neutrophils’ degradation via lysis [4].

Leukocidins are pore-forming cytotoxins that help the bacteria invade host cells. The bi-component pore-forming leukocidins (Luk) include LukAB and lukED. LukAB is a surface-associated molecule secreted outside the bacterial cell that leads to lysis of host immune cells. LukED, similarly to γ-haemolysin CB (HlgCB), leads to neutrophil lysis. LukAB and LukED lead to extended attacks on macrophages [5].

In staphylococci, most virulence factors, including surface proteins and toxins, are regulated by transcriptional regulators’ two-component systems (TCSs) and quorum sensing systems (QS) with an accessory gene regulator (*agr*) [6].

Strains may contain one of four classes of accessory gene regulator (*agr*) locus (on the basis of polymorphism of *agrD* and *agrC* genes) marked as *agrI*–*agrIV*. The *agr* regulation system belongs its effector, called RNAIII. It affects the expression of many virulence factors, e.g., exotoxins, sae, and genes involved in biofilm formation, peptidoglycan and amino acid metabolism, as well as transport pathways [7]. It also encodes for δ-haemolysin and therefore the expression of *hld* serves as a surrogate marker to assess agr functionality [8]. Another element of regulation of the expression of virulence toxins is the *S. aureus* exoprotein expression (Sae) locus. Together with *agr,* it activates production of toxins, thus affecting bacterial virulence [9].

Interestingly, *agr* genes together with *RNAIII* positively regulate genes that are expressed postexponentially, while coagulase is inhibited, as its expression may be control positively and negatively by the *agr* system according to the growth stage [10].

The main problem with staphylococci eradication is their growing resistance to antibiotics. Most genes involved in antibiotic resistance are localized on plasmids, and thus can be transferred from one strain to another quite easily. The most common *S. aureus* resistance is mediated by the *mecA* gene element, found in the mobile genetic element—the *SCCmec*. It can be transferred from one strain to another through horizontal gene transfer, and thus may be detected in various CoNS as well as coagulase-positive staphylococci (CoPS) species, but in *S. sciuri* and *S. vitulinus mecA* alleles do not lead to resistance to beta-lactams. The *mecA* gene encodes PBP2′ (PBP2a), which is responsible for resistance to methicillin with cross-resistance to other drugs from the β-lactam group [11].

The detection and spread of multidrug-resistant (MDR) CoNS in hospital settings is a rising problem. For instance, *S. epidermidis*, *S. haemolyticus*, and *S. hominis* are some of the most prevalent factors in infective endocarditis, bloodstream infections (BSI), and neonatal sepsis in neonatal intensive care units (NICU) [12]. Studies from 49 hospitals in the United States of America indicated that CoNS represented 31% among all cases of nosocomial BSI within a period of 7 years [13], while studies from Germany confirmed CoNS as the second most common factor of nosocomial infections [14]. Additionally, infections caused by CoNS are particularly common in newborns and preterm neonates when compared to healthy children or adults [15,16]. Besides a high frequency of resistant CoNS in hospitals, the frequency of drug resistance of CoNS isolates from the environment is also growing (including resistance to last-resort antibiotics, e.g., oxazolidinones and lipopeptides) [3,17]. It is noteworthy that according to the literature, the prevalence of MRCoNS is lower than that of MRSA in most studies, but it might be a result of the fact that occurrence of infection with CoNS is not as common as that with *S. aureus* and often CoNS are not properly diagnosed as the factor of disease, considered only as a commensal members of the skin microbiota [18]. Anyway, the evolution of CoNS from commensals into invasive, resistant pathogens is happening [19].

As our initial, already published, studies [20] indicated that some tested CoNS isolates contain virulence factors that originally come from *S. aureus*, we decided to investigate this phenomenon further. Additionally, the aim of the present studies was to check if clinical CoNS strains that contain chosen virulence factors were also enriched with elements responsible for regulation of genes that encode virulence factors typical for *S. aureus*. Moreover, we checked weather these regulation factors can affect virulence activity of other strains. For this purpose, we decided to screen some more isolates of *S. haemolyticus*, *S. hominis*, *S. simulans*, *S. warneri,* and additionally *S. epidermidis* species that were confirmed as etiological factors of chosen human infections, for presence of virulence factors to find more strains with different combinations of virulence factors and elements involved in cell regulation required for cross-talk studies. For these purposes, we tested phenotypically as well as genotypically the presence and activity of virulence factors that are homologous to *S. aureus*. In these studies we also analyzed the association between methicillin resistance and virulence factors of CoNS isolates, as in *S. aureus* strains a significant relationship between antibiotic resistance and haaemolysin genes has been observed.

The knowledge related to the presence and regulation of virulence factors typical for *S. aureus* in CoNS strains may help to understand their less known, pathogenic nature, and thus help to visualize the rising problem of CoNS as an etiological problem of infection but also find a way to eradicate them.

## 2. Materials and Methods

### 2.1. Material Isolation and Identification

#### 2.1.1. Tested Strains

In current studies, 47 new, not presented before, CoNS isolates were analyzed. Among them, 6 isolates were identified as *S. haemolyticus*, 6 as *S. hominis*, 5 as *S. simulans*, 6 as *S. warneri,* and 6 as *S. epidermidis* by MALDI-TOFF and genetic analysis [21]. Additionally, 19 isolates that were partially introduced in a previous manuscript [20] were further investigated. Among tested isolates, 4 of them were identified as *S. haemolyticus*, 8 as *S. hominis*, 4 as *S. simulans*, and 3 as *S. warneri*. All isolates were from human blood, wounds, the peritoneum, urethra, and skin dermatitis, or in some cases from other parts of the body, such as eye or appendix.

As controls, the following strains were used: *S. haemolyticus* ATCC 29970, *S. hominis subsp. hominis* ATCC 27844, *S. warneri* ATCC 27836, and *S. simulans* ATCC 27848.

All isolates used were acquired from a diagnostic microbiology laboratory (Synevo Sp. z o.o.) of Lodz area, Poland, during 2014–2016.

#### 2.1.2. DNA and RNA Isolation

The DNA and RNA materials were isolated using the Genomic Micro AX Staphylococcus Gravity and Total RNA Prep Plus Minicolumn Kit (A&A Biotechnology, Gdańsk, Poland), respectively, according to the protocol provided by the manufacturer. The DNA and RNA quality and concentration were evaluated using the Spectrophotometer Genova NanoJenway.

### 2.2. Analysis of the Virulence

#### 2.2.1. Phenotypic Analysis of Haemolytic Activity of Tested Isolates

The phenotypic manifestation of haemolysines activity was determined on an agar medium with 5% sheep blood. Additionally the activity of β-toxin was tested by reverse CAMP test as well as analysis of hot–cold effect. Haemolytic activity of β-toxin is promoted after incubation at temperatures below 10 °C, thus this toxin is often called as the ‘hot-cold’ haemolysin. Production of δ-toxin was detected by the presence of synergism with β-haemolysin of *S. aureus* ATCC 25923 (CAMP test). δ-haemolysin produced by a test strain enhances the β-haemolysis caused by *S. aureus* ATCC 25923 strain. Briefly, tested isolates were streaked perpendicularly to an *S. aureus* ATCC 25923 strain. Plates were incubated at 37 °C for 18 h [22].

#### 2.2.2. Phenotypic Analysis of Biofilm Formation

Biofilm formation assay was performed as previously described [23]. CoNS cultures were grown for 24 h at 37 °C with agitation. Next, isolates were grown to the turbidity of a 0.5 McFarland standard in TSB medium and 100 μL of each dilution was loaded into the wells of a non-treated flat-bottom 96-well microtiter plate (Nunc). *S. aureus* ATCC6538 (biofilm forming) and *S. epidermidis* ATCC12258 (not biofilm-forming) were used as controls. After 24 h incubation at 37 °C for biofilm production, the supernatants were removed and the adherent cells were stained with an aqueous solution of crystal violet (0.1%, *w/v*) at room temperature and 2× washed with distilled water. Then, the microtiter plates were dried for a few hours. Bound crystal violet was dissolved by treatment with 30% acetic acid at room temperature and optical density (OD) of each well was measured by using a microplate reader (Biotec) at 550 nm. Wells that contained broth only were used as negative control.

Biofilm density was classified according to the scheme presented by Stepanovic et al. [24]. Three standard deviations above the mean OD of the negative control were calculated as the cutoff value (ODc) for each microtiter plate. According to ODc and average OD of the strain, isolates were described as: strong biofilm producer (4ODc  ≤  OD); moderate biofilm producer (2ODc  ≤  OD  ≤  4ODc); weak biofilm producer (ODc  ≤  OD  ≤  2ODc); and no biofilm producer (OD  ≤  ODc). Each test was repeated at least four times.

#### 2.2.3. Detection of Genes Encoded Virulence Factors

CoNS isolates (30), not presented previously, were screened for the presence of the following virulence-associated genes: 1. Involved in biofilm formation, *icaA, icaB, icaC,* and *icaD*; 2. Panton–Valentine leukocidin, *pvl*; 3. staphylococcal enterotoxins, *sea, seb, sei seg*.; 4. haemolysins: *hla, hlb, hld*, γ-haemolysin component A (*hlgA*), γ-haemolysin component B (*hlgB*). *S. aureus* ATCC25923 and *S. epidermidis* ATCC12228 strains were used as positive and negative controls. PCR reactions were performed using the primers and parameters described in Table 1. 

Additionally all 49 tested strains were screened for the presence of virulence-associated leucotoxins *lukAB* (636bp) and *lukCD* (269bp), and *hlgCB* genes according to the literature listed in Table 1. The primer sequences for genes encoding the virulence factors are listed in Table 1.

### 2.3. Antimicrobial Susceptibility Testing

Methicillin-resistant strains were screened by the disc-diffusion method (using Becton Dickinson discs, cefoxitin FOX-30) in accordance with the EUCAST guidelines. *S. aureus* ATCC 29213 was used as a control strain.

Polymerase chain reaction (PCR) was executed according to the literature [28]. *S. aureus* 51625 strain was used as a control.

### 2.4. Agr Detection and Classification

Agr activity of tested isolates was analyzed by the assessment of δ-haemolysin production. Production of δ-toxin was detected as described above [22]. 

The classification of *agr* system groups was based on the hypervariable domain of *agr* locus according to Soares et al. [29]. Multiplex PCR test was performed to type groups based on their product size according to the literature presented in Table 2. The primers sequences for genes encoding the factors involved in regulation of virulence are listed in Table 2.

The genes’ expression was tested by a real-time PCR assay on the Rotor-Gene™ 6000 thermocycler (Corbett Life Science; Qiagen). The primer sequences of used primers are presented in Table 3. Briefly, PCRs were set up using cDNA derived from the input RNA. Reverse transcriptase (RT) reaction was performed using *Enhanced Avian HS RT-PCR* Kit (Sigma) in accordance with the manufacturer’s protocol. Gene DHFR encoded dihydrofolate reductase was used as a reference.

### 2.5. Cross-Inhibition of the Biofilm Formation and the Influence on δ-Toxin Activity among Agr Groups

#### 2.5.1. Supernatant Preparation

The single colonies of CoNS strains with functional *agrI*, *II*, or *III* were grown overnight in 4 mL of TSB (Becton Dickinson) with 100 rpm agitation at 37 °C. After the incubation time, 50 mL of new TSB medium was inoculated with the overnight cultures of each strain and incubated at 37 °C with 100 rpm agitation for 18 h. Next, supernatants were obtained by centrifugation, and cells were eliminated by filtration through a 0.2 μm membrane. The supernatants were stored at 4 °C. The microbiological purity of supernatants was tested on agar medium with 5% sheep blood [44].

#### 2.5.2. Analysis of Cross-Inhibition of Biofilm Formation

CoNS strains with not functional *agrI*, *agr II,* or *agr III* genes were grown for 24 h at 37 °C with agitation. Next, isolates were grown to the turbidity of a 0.5 McFarland standard in TSB medium and 100 μL of each sample was loaded into the wells of a non-treated flat-bottom 96-well microtiter plate (Nunc); then, 100 μL of supernatant was added. The CoNS strains with non-functional *agrI*, *II*, or *III* genes were grown in this TSB supernatant (50–50%) mixture at 37 °C for 24 h. Next, the determination of biofilm formation was executed as mentioned above. *S. aureus* ATCC6538 (biofilm-forming) and *S. epidermidis* ATCC12258 (not biofilm-forming) were used as controls [44].

#### 2.5.3. Analysis of the Influence on δ-Toxin Activity

CoNS strains with non-functional *agrI*, *II*, or *III* genes were grown for 24 h at 37 °C with agitation in TSB supernatants (50–50%) mixture. Production of δ-toxin was assessed by CAMP test as described above [22].

### 2.6. Statistical analysis

To assess correlation of variables, the phi square test was used. The results in phi tests range from 0 to 1, where 1 means a significant association, while 0 means no relationship; thus, results can be interpreted as follows: >0—no or very weak, > 0.05 weak; > 0.10 moderate; > 0.15 strong; > 0.25 very strong.

## 3. Results

### 3.1. The Presence of Virulence Factors, mecA Gene, and Susceptibility to Methicillin

The results of phenotypic manifestation of virulence factors and the presence of their genes in CoNS isolates are presented in Figure 1.

Most *S. haemolyticus* and *S. simulans* isolates indicted haemolytic activity on agar plates supplemented with sheep blood and gave a positive result of the CAMP, which proves that they contained active Β and δ haemolysin genes. Similar haemolytic properties were presented in at least half of tested *S. epidermidis* and *S. hominis* isolates, but just a few *S. warneri* isolates. Moreover, most *S. homonis* isolates indicated positive CAMP results, while just some isolates from other species presented δ haemolysin activity. None of tested isolates gave a positive result of the CAMP reversed test.

Interestingly, at the same time, the *hla, hlb* genes that encode haemolysins were not detected in any of the tested isolates, while *hld* was identified in only one of the tested *S. haemolyticus* isolates. The *hlgA* component was detected in less than a quarter of *S. hominis* isolates but subunit *hlgB* was not detected in any tested isolates. The *hlgCB* unit was identified with similar frequency in *S. simulans* isolates and even less often in *S. hominis* isolates. The most common gene among all detected genes was *lukAB,* detected in isolates belonging to all tested species. It was the most prevalent in *S. haemolyticus* and *S. warneri* isolates, while in other isolates beside *S. epidermidis*, its frequency was not lower than 30%.

Most *S. epidermidis*, *S. haemolyticus,* and *S. homonis* isolates were able to produce biofilm, but among all tested elements of *icaADBC operon* only the *icaA* gene was detected in three out of five tested species; *the icaR* transcriptional repressor was found only in a few *S. haemolyticus* isolates, while other genes were not detected.

Neither the *pvl* nor any of the enterotoxins genes were detected in any of the tested isolates.

All tested *S. hominis* and most *S. haemolyticus* isolates were resistant to methicillin, while all *S. warneri* isolates indicated sensitivity to the mentioned antibiotic. Phenotypically obtained results were in accordance with results obtained from *mecA* gene detection studies, as the *mecA* gene was not detected in any *S. warneri* isolates, while it was especially common in *S. hominis* and *S. haemolyticus* isolates. The *mecA* gen was not detected in isolates classified as *S. simulans*, while a few of these isolates were resistant to methicillin in phenotypic studies.

### 3.2. The Presence of Regulatory Factors

As presented in Figure 2A, among genes that belong to the accessory gene regulatory (*agr*) system, *agrI* was detected in isolates assigned to four out of five tested species, but in all cases with quite moderate frequency. The *agrII* gene was present in more than a quarter of *S. simulans* isolates and just a few *S. warneri* isolates. The *agrIII* gene was found in only several *S. epidermidis* isolates, while *agrIV* was not detected in any tested isolates. Other tested genes involved in regulation of the virulence of bacteria were quite common in tested species; *sar* and *sae* genes were found in isolates that belonged to all tested species except *S. simulans* isolates or *S. hominis and S. haemolyticus* isolates, respectively. The *RNAIII* gene was not found in any *S. warneri* isolates, while in isolates that belonged to other species its frequency was quite moderate.

### 3.3. Statistical Analysis Materials: Będzie Ficant Associa

Statistically, a significant association between the presence of the *agrI* gene and *lukAB, lukED, icaR, and hlgCB* genes for all tested isolates from all tested species was observed (for all tested isolates *p* value = 0.0015–0.04), but at the same time no association between the presence of the *agrI* gene and any phenotypically demonstrated features was observed (for all tested isolates *p* value = 0.12–0.7).

The meaningful relationship between the presence of *agrII* and *hlgA*/*lukAB* genes was found for isolates classified as *S. simulans* (*p* value = 0.0027 in both cases).

Significant association was also observed for the *agrIII* gene and *lukAB*, *lukED*, and *icaR* genes in isolates classified as *S. epidermidis* (*p* value = 0.014 in all cases).

Interestingly, the association between particular *agr* genes, the *mecA* gene, and phenotypically manifested virulence features was observed only between *agrII* genes and biofilm formation for *S. simulans* isolates, the *agrII* gene and results of the CAMP test for *S. warneri* isolates, the *mecA* gene and β-haemolysis for *S. haemolyticus,* and *mecA* and the result of the CAMP test for *S. hominis* isolates.

There was also no meaningful correlation between antibiotic resistance manifested phenotypically and the presence of the genes for alpha, delta, and beta haemolysins.

There was no association between the presence of the *mecA* gene and any genes responsible for virulence or genes involved in regulation of these genes. On the other hand, the association between the presence of the *mecA* gene and biofilm formation manifested phenotypically (for *S. haemolyticus* isolates), the mecA gene and β-haemolysis (for *S. hominis* isolates) was observed.

### 3.4. Activity of Genes Encoding Virulence and Regulatory Factors

As presented in Figure 1C, the *hlgA* component presence was detected only in several *S. hominis* isolates. It has to be noted that activity of the *hlgAB* gene requires the presence of either active *hlgA* or *hlgB* components; thus, as the *hlgB* gene was not found in any tested isolates, the *hlgAB* gene was not active in any tested isolates. The expression of *hlgCB* and *lukED* was confirmed in all *S. simulans* and *S. hominis* as well as *S. haemolyticus* and *S. hominis* isolates, respectively. The functional *lukAB* complex was detected in less than half of *S. hominis* isolates enriched with this gene. The *hld* gene was not active in tested isolates. The frequency of isolates with an active *agrI* gene was marginal in *S. hominis* and *S. haemolyticus*, while most *agrII* genes present in *S. simulans* isolates were expressed (see Figure 2B). The *agrIII* genes were active in all *S. epidermidis* isolates. The *sae* gene did not exhibit expression in any tested isolates, while *sar* gene expression was detected in less than a quarter of *S. warneri* isolates. A few more *S. simulans* isolates indicted the *RNAIII* expression.

It is noteworthy that in only one isolate that was classified as *S. simulans,* species *agrI* gene and *hlgCB* genes were expressed, while *lukAB* gene was inactive. All isolates with an active *agr* gene (*agrI* or *agrIII*) carried the inactive *lukAB* gene.

Among all 49 tested isolates, 11 expressed one of the *agr* genes; 27% of these isolates came from blood, 45% from wounds, and 18% from foot ulceration. Among 7 of the most expressed isolates (that contained at least 3 genes involved in bacterial virulence), 43% came from blood and 28% came from peritoneum. Among *S. epidermidis* isolates, only one of them contained inactive *agrI* and active *agrIII* genes at the same time, as well as inactive *lukAB, sae, sar*, and *mec* genes. It gave a negative result for the CAMP test and to β-type haemolysis. All *S. hominis* isolates with active *agrI* genes led to β-type haemolysis and gave positive results for the CAMP test. Additionally, one of these isolates indicated inactive *hlgA* and *hlgCB* genes, but an active *mecA* gene. All *S. simulans* isolates with active *agrII* genes led to β-type haemolysis and gave positive results for the CAMP test, but in none of them was the *hld* gene detected. Only one isolate classified as *S. haemolyticus* with an active *agrI* gene was detected. This isolate had also *hld, lukED,* and *RNAIII* genes, but none of these genes was active and none of these isolates led to β-type haemolysis and gave positive results for the CAMP test.

### 3.5. The Cross-Talk Analysis

To check whether the inter- and intra-strain and species cross-inhibition (interaction) may occur, isolates with inactive *agr* genes were incubated with cell-free supernatants from strains with active *agr* genes. The influence of one isolate on another one was analyzed by phenotypic manifestation of biofilm formation and activity of δ-toxin in the CAMP test (obtained results are presented in Table 4).

It has been observed that supernatants led to reduction of biofilm formation by isolates with inactive *agr* genes, except the *S. epidermidis* ATCC12228 strain, as in this case used supernatants did not affect biofilm production. No changes in the level of inhibition between supernatants from different isolates were observed and there was no difference whether: (1). Supernatants came from isolate of the same species as treated isolates or not, (2). Supernatants came from isolates with strong or weak ability to form biofilm, (3). Treated isolates were strong or weak biofilm producers. As supernatants did not affect bacterial growth, it means that reduction of biofilm formation by tested supernatants was not a result of changes in bacterial growth.

All used supernatants from isolates with an active *agrII* gene or combination of inactive *agrI* and active *agrIII* gene were able to activate previously inactive δ-haemolysin activity of isolates with inactive *agr* genes, thus giving positive results in the CAMP test. Incubation of all tested isolates with supernatants from *S. hominis* 01 and *S. hominis* 02 (with active *agrI* gene) isolates led to negative results in the CAMP test, while incubation of all tested isolates with supernatant from *S. haemolyticus* (with active *agrI* gene) isolates led to positive results in the CAMP test for all tested isolates. Similarly to results from the test of the influence of supernatant on biofilm formation, no differences were observed when supernatants came from the isolate that belonged to the same species as treated isolate.

## 4. Discussion

Virulence factors including hydrolytic enzymes, leukocidin, enterotoxins, and haemolysins play important roles in staphylococcal escape from both innate and adaptive immune responses, and growth and spread of bacteria in the host. The presence of virulence factors in CoNS may not only increase their pathogenicity, but also make them a reservoir for resistant genes that can be transmitted to other pathogens [45].

Haemolysins produced by staphylococci indicate cytolytic effects on many different types of cells, e.g., erythrocytes, platelets, monocytes, and neutrophils [46]. During chronic pathogenesis, the selective pressures from antibiotic activity and the host immune response can increase expression of α-haemolysin, which is involved in colonization of *S. aureus* in respiratory tract infections. Beta-haemolysin (Hlb), besides induction of haemolysis on blood agar plates in a limited number of *S. aureus* strains, increases the host cell susceptibility to α-haemolysin and PVL and leads to lymphotoxicity, and DNA cleavage [47,48]. The *hld* gene that encodes delta haemolysin is located within the *RNAIII* locus of accessory gene regulator (*agr*), thus the expression of delta haemolysin genes can be a useful marker of *agr* function [49]. Delta haemolysin activates neutrophils leading to the generation of reactive oxygen intermediates (ROI) and induces the release of pro-inflammatory cytokines from keratinocytes [50]. It may lead to intestinal diseases, e.g., acute diarrhea to severe enteritis [51].

δ-Haemolysin in opposition to Γ-haemolysin, leads to destruction of red blood cells only at high concentrations as it is able to create a trans-membrane pore which lyses the cell membrane. It is well documented that HlgAB and HlgCB toxins require the presence of a class S and a class F component for increased activity and the relation between these two elements is synergistic. [52]. Overexpression of HlgAB determines the highly virulent phenotype of Newman strain as it is strongly responsible for haemolytic activity of bacterial strains. Interestingly, in the presence of HlgAB toxin, the *hla* gene could indicate only marginal activity [53].

In our studies, the *hlgB* gene was not found in any tested isolates, which suggests that the *hlgAB* complex was inactive in all cases. It has to be noted that γ-haemolysin components are located in the core genome and are highly conserved in most *S. aureus* lineages, but the diversity within their coding sequences had been proved [54].

Although *lukAB* genes, similarly to *hlg* genes, are located on core genome, and thus should not be as easy a subject of horizontal transfer as genes located on plasmids or pathogenicity islands, they were the most commonly detected in tested isolates. In contrast to *lukAB*, the *lukED* gene sequence is highly conserved [55] and unlike *lukAB*, and *hlg*, is located on pathogenicity island (SaPI) vSa, which is stable and not relocated via horizontal gene transfer [56]. Additionally, sequence identity between LukED and PVL is around 75% [37]. In our studies, the *lukED* gene was also detected in *S. epidermidis*, *S. hominis,* and *S. haemolyticus* isolates, but with lower prevalence than *lukAB*, although higher than *pvl* gene, and its activity was confirmed in all tested *S. hominis* and *S. haemolyticus* isolates. According to the literature, the prevalence of the lukED genes may vary between different strains as in some studies lukED genes were found in 87% of *S. aureus* strains while in other studies a prevalence of about 30% in human clinical and colonizing isolates was noted [37,57].

The *hla, hlb* genes were not detected in any tested isolates, although most *S. haemolyticus* and *S. simulans* isolates led to haemolysis on agar plates with sheep blood. This means that probably most tested isolates contained genes encoding B haemolysin with a different sequence than that used in our studies’ sequence from *S. aureus*. Additionally, neither *pvl* nor any enterotoxins genes were detected in tested isolates, while the *hld* gene was detected only in one *S. hominis* isolate. The obtained results are in accordance with our already published results of the characteristic of CoNS isolates from, e.g., blood and wounds, and allow us to conclude that transfer of *hla, hlb,* as well as *pvl* and any tested enterotoxins genes from *S. aureus* into tested CoNS isolates via horizontal transfer did not occur [20]. It is well documented that some toxins, e.g., alpha haemolysin, have been encoded in the genome, while others, e.g., enterotoxins, Panton–Valentine leukocidin (PVL), are encoded on mobile genetic elements such as plasmids, prophages, transposons, or pathogenicity islands [58,59]. As mobile genetic elements are effective vehicles for spreading virulence and drug-resistant genes between *S. aureus* strains through horizontal gene transfer, we should expect that toxin genes such as *sea*, *pvl* should be acquired by tested isolates more easily than *hlgAB* or *hlgCB* genes localized on the core genome, but the obtained result show the opposite tendency. Moreover, studies presented by some other researchers indicated the prevalence of these genes in CoNS isolates. For instance, Udo et al. indicated that the *seb* gene was dominant in CoNS isolated from adult patients [60], while in studies presented by Vasconcelos et al., gene was the most common among tested virulence factors in CoNS isolates obtained from newborns in Brazil [61]. Nasaj et al. indicated that more than 50% of analyzed CoNS isolates from hospitalized patients in Iran showed prevalence of *hla, hlb, hlg, and hld* genes [62]. Nevertheless, there are suggestions that the presence of virulence genes in CoNS isolates may vary depending on the geographical locations [60,62,63] and different environmental conditions may cause extensive changes in resistance and pathogenicity of bacteria [64,65].

Many CoNS are able to form biofilms to enhance their virulence as well as to protect themselves from the diffusion of antibiotics into the host cells [66]. Biofilms are able to adhere to various objects, e.g., indwelling medical devices. This process is possible due to surface proteins’ polysaccharide intercellular adhesion (PIA), regulated by *icaADBC* genes [67]. The most prevalent among *ica* genes was *icaA,* detected in *S. epidermidis*, *S. hominis,* and *S. haemolyticus* isolates. Besides *icaA*, only the *icaR* gene was found in some *S. haemolyticus* isolates and no other *ica* genes were disclosed in tested isolates. Anyway, most tested isolates were classified phenotypically as biofilm producers. Kord et.al. proved that bacterial strains are able to form biofilms in the absence of *ica* genes, which may suggest the presence of specific mechanisms of biofilm formation independent of *ica* genes [68]. Interestingly, recent studies have indicated that haemolysins are also involved in biofilm formation in *S. aureus* [69].

Antibiotic resistance of staphylococci including CoNS, especially against betalactams over the years [70]. It is well documented that when methicillin-sensitive *S. aureus* (MSSA) strains receive *SCCmec* complex, they become resistant to methicillin [71]. Nowadays, the presence of the *mecA* gene is quite common in staphylococci. For instance, it has been detected in 80% of the CoNS isolates causing late-onset sepsis in neonates [69]. Additionally, some strains may contain also another *mec* gene, a *SCCmec* element type XI called *mecC* [72]. The *mecC* gene is not as common in staphylococcal strains as the *mecA* gene, but slowly the frequency of its presence in staphylococcal strains has become ubiquitous. In our studies, although the *mecA* gen was not detected in any isolates classified as *S. simulans* and in only around 30% *of S. haemolyticus* isolates, respectively, almost a quarter of all these isolates were resistant to methicillin in phenotypic studies. The obtained results suggest that these isolates may harbor the *mecC* gene, which is responsible for resistance to cefoxitin, giving in our case MRSS and MRSH phenotypes, respectively [73]. Moreover, only around 50% of *S. haemolyticus* isolates with the *mecA* gene present indicated resistance to methicillin phenotypically, which suggests that in only around half of these isolates was the *mecA* gene active. It should be noted that the obtained results come from a limited number of isolates classified as particular species but also that many different factors may lead to antibiotic resistance, including age, gender, climatic conditions, food type, and especially regional culture [74]. According to the literature, prevalence of methicillin-resistant CoNS may vary depending on the geographic regions, with MR-CoNS rates ranging between 16 and 50% [75,76]. Community-acquired commensal CoNS in Europe indicated low prevalence of *mecA*. For instance, in studies that were carried out in Germany, *mecA* detection among the CoNS isolates was low (i.e., 7%) and only 10% of tested patients were colonized by at least one MR-CoNS isolate. Studies from Etiopia confirmed high frequency (17%) of the presence of resistant CoNS strains (more than 50% were resistant to penicillin, ampicillin, tetracycline, or sulfamethoxazole-trimethoprim) [17,77,78].

The staphylococcal *agr*-QS system upregulates α, β, γ, and δ–haemolysins, leukotoxins, lipases, while it represses the transcription of cell wall-associated proteins [79], thus the loss of activity of *agr* genes reduces expression of genes encoded haemolytic activity and the isolate without functional *agr* should produce only one type of haemolysin (usually a-haemolysin) [80]. This system also coordinates the expression of its effector molecule RNAIII, which as was already mentioned modulates the expression of virulence factors at transcriptional and post-transcriptional levels [81]. There are suggestions that enterotoxins and leukocidins are upregulated also by *sae* (accessory element) and *sar* [82]. Interestingly, the *agr* system upregulates toxin genes’ expression during the late phase of growth and downregulates the cell surface factor by responding to auto-inducing peptides (AIP) produced by *S. aureus* [81].

In our studies isolates from wounds and blood showed the most prevalence of *agr* genes; isolates from peritoneum were also well equipped with genes from the *agr* regulatory system. Li et al. indicated that *S. epidermidis* isolates from catheters, blood cultures, urine, wounds, sputa, cerebrospinal fluid, and dialysate were prevalent in *agr* group I genes, while *agr* group III genes were rarely detected [83]. Our analysis, based on polymorphisms of the *agr* gene, indicated that *S. epidermidis, S. hominis,* and *S. haemolyticus* isolates were prevalent in *agrI* gene similarly to MRSA isolates from many already published studies [84,85], while *S. simulans* and *S warneri* isolates contained mainly *agrII* genes. The *agrIII* genes were detected only in some *S. epidermidis* isolates, but all of these genes were active. Moreover, *S. simulans* isolates were the most frequently equipped with active *agr* genes (55% of tested isolates in total), while in isolates denoted as other CoNS species, the presence of *agr* genes was not very common. It is postulated that agr group I is strongly associated with CA-MRSA genotypes, while agr group II is more correlated with HA-MRSA in human isolates [86]. In addition, another study reported that methicillin resistance of bovine isolates is more prevalent in agr group I than other groups [87]. The mecA genes are found in agr groups I, II, and III, but group IV strains have not acquired a SCCmec element [88].

Beside *agr* genes, the most frequently detected gene responsible for regulation of virulence was the *sar* gene, but its activity was limited to only a few isolates denoted as *S. simulans* and *S. warneri*. The presence of *sae* and *RNAIII* genes was rather marginal and in only a few cases were RNAIII genes active, while in all tested isolates enriched in the *sae* gene it was inactive. These genes were detected only in *S. epidermidis*, *S. simulans,* and *S. warneri* isolates. *S. warneri* isolates and *S. hominis* isolates were enriched in the *sar* gene, while in isolates assigned to other CoNS species we detected similarly marginal frequency of the *sae* gene. The *agrII* genes in most *S. simulans* and all *agrIII* genes in *S epidermidis* isolates were active, while only in some *S. simulans* and *S. warneri* isolates were active *RNAIII* and *sar* genes found. This means that in any of tested isolates, the complete set of functional *agr* regulatory system homologous to *S. aureus* was not found. It is noteworthy that the identity in the *agr* locus among different *Staphylococcus spp.* was high, especially in the first 50 and last 150 nucleotides, but also many insertion/deletions, especially in the 5’ end may be found [89]. For instance, the *agr* locus of *S. epidermidis* is nearly 68% homologous to *S. aureus* locus [90].

It is postulated that superantigens production in *S. aureus* are directly correlated to the *agr* type of isolates. Many studies presented in the literature indicated that isolates prevalent in superantigens usually contained *agr* type I or III genes, which is in accordance with our studies where isolates with detected toxin genes were prevalent in the *agrI* gene. However, in some other reports, *agrII* isolates were predominant. Additionally, significant association between many virulence factors and *agrI* was indicated, but biofilm, *hlb, hlg* and *hld* were significantly associated with *agrIII*. Other studies also confirmed that staphylococcal strains enriched in *agrIII* and *agrII* more often produce biofilm than strains contain other types of agr [91,92,93], while in our studies most isolates enriched with genes encoding haemolysins or leukocidines belonged to *agrI*.

In nature, most microorganisms grow in multispecies communities where inter- and intraspecies relations take a place. For instance, Todd et.al. indicated that *C. albicans* when grown with *S. aureus* shows effects its virulence factors’ protein expression [94]. Cross-communication involving *agr* interference between particular strains of *S. aureus* and chosen other species, e.g., *S. epidermidis* and *P. aeruginosa,* within the same ecological niche has previously been observed. Additionally, some staphylococcal isolates belonging to *S. epidermidis* species via autoinducing peptide (AIP) molecules inhibit *agr* activity of *S. aureus* groups I, II and III, but not IV, while only AIP secreted by *S. aureus* with active *agrIV* gene are able to inhibit *S. epidermidis agr* activity [95]. It is postulated that the intra- and inter-cross-talk via the agr quorum sensing system may affect the virulence as well as colonization ability of staphylococci similarly to other bacteria [96]. In studies described by Martínez-García et al., *AgrI* and *AgrII* culture supernatants from *S. epidermidis* significantly reduced the biofilm formation of *AgrI*, *II*, and *III S. epidermidis* strains, while the *AgrIII* supernatant did not affect the biofilm formation of the *AgrII* strain [44]. Canovas et al. also proved that staphylococci, e.g., *S. schleiferi* from varying environmental niches, can affect virulence and colonization of *S. aureus* [97]. The observation was that *S. schleiferi* is a potent inhibitor of *S. aureus agr genes* and from all *agr* types (from I to IV). It offers a potential new avenue for exploring staphylococci as sources for quorum sensing inhibitors that can be used to target *S. aureus agr*-related infections. *S. schleiferi* is also able to inhibit *S. aureus* strains by affecting its *agr* gen activity, especially strains with the *agrI* gene [98]. 

Although the *agr* cross-inhibition between the different *S. aureus agr* groups and between *S. aureus* and some other staphylococcal species (especially *S. epidermidis*) is quite well documented, little is known about the relations between CoNS species, especially about the influence of *agr* genes on the activity of virulence factors [33,99]. Our results indicated that CoNS isolates, similarly to *S. aureus* strains with active *agr* genes, are able to inhibit biofilm formation of other CoNS isolates with inactive *agr* genes and there was no difference whether isolates contained the same type of *agr* or not. Moreover, there was no distinction in the obtained results as to whether isolates were assigned to the same or different species, which means that there was no cross-species barrier for *agr* system activity. Our findings are in agreement with data provided by other groups; it is well documented that there is downregulation of agr facilitate biofilm development in staphylococci [100,101]. The agr-deficient variants produce much higher but worse-organized biofilm mass when compared to strains with active *agr* genes. This might be related to the fact that the lack of agr-dependent production of PSMs involved in functional biofilm architecture is dependent on the agr system [102]. The level of biofilm production was changed only for S. epidermidis ATCC 12228 strain. This might be related to the fact that the *S. epidermidis ATCC12228* strain is generally a weak biofilm producer.

It is postulated that an active agr system is necessary to induce synergistic haemolysis that can be observed by the CAMP test [103]. Our studies all used supernatants from isolates with active *agrII* genes, or a combination of inactive *agrI* and active *agrIII* genes were able to activate previously inactive δ-haemolysin activity of isolates with inactive *agr* genes, suggesting that strains with active *agrII* and *III genes* are able to affect activity of virulence factors of other strains. A more complex situation was observed when isolates with inactive *agrI* genes were incubated with supernatants from isolates with active *agrI* genes as supernatant from some isolates led to inhibition of δ-haemolysin (and negative results of the CAMP test) of some isolates, while others activated expression of inactive δ-haemolysin genes in other tested isolates. This might be a result of the fact that Hld is encoded by RNAIII, the effector of agr [32], thus the virulence factor of isolates with active RNAIII gene (*S. simulans 02* and *S. epidermidis ATCC 12228)* might be muted by Agr from supernatant from isolates with active *agrI* genes, while virulence genes from isolates with inactive RNAIII genes (*S. simulans 01*, *S. epidermidis*) will not be affected by Agr from these supernatants. On the other hand, incubation of all tested isolates with supernatant from isolate without RNAIII gene (*S. haemolyticus)* isolates led to positive results of CAMP tests for all tested isolates. The obtained result might be associated with the fact that supernatants from both used *S. hominis* isolates also contained active *mecA* genes, while *S. haemolyticus* isolate was sensitive to methicillin phenotypically and the presence of *the mecA* gene was not found. Additionally, one of these isolates indicated an inactive *hld* gene, but an active *mecA* gene. It has been proved that in MRSA strains, the *mecA* gene leads to some changes in virulence factors of the organism [104]. The activity of this gene affects some proteins such as *agr*, and *agr*-regulated enterotoxins and PVL [105]. This finding is in accordance with our results as all *S. hominis* isolates with active *agrI* genes led to β-type haemolysis and gave positive results in the CAMP test.

There are suggestions that anti-virulence therapy and treatment targeting the *agr* system might be a good alternative to common antibiotics in antimicrobial therapy [106].

Bacterial adaptation to a changing environment is often associated with the mutation in genes involved in their virulence. For instance, it is postulated that staphylococcal resistant to methicillin with *agr* variations due to their defects in activity of *agr*-depending quorum sensing system may contain additional agr-independent virulence factors as a likely part of their survival strategy. This strategy may lead to pleiotropic effects and the generation of phenotypic heterogeneity [107], enabling success of certain *S. aureus* strains, e.g., the CA-MRSA USA300 clone that strongly expresses toxins under control of the *agr* system, thus inducing itchy skin infections, and is pandemic in the USA [108,109].

## 5. Conclusions

Our results, similarly to already published data, indicate that CoNS may soon become significant pathogens. The variety of properties among isolates, which is the result of HGT, confirms the formation and selection of methicillin-resistant strains as well as strains with new combinations of virulence features among CoNS. Our results confirmed the intra- and inter-cross-talk between chosen staphylococcal isolates mediated via the *agr* quorum sensing system and suggest that the *agr* quorum sensing system may play an important role for inter-species communication and that this may be involved in the changes in the virulence of staphylococci. These studies broaden our understanding of antibiotic resistance and carriage of virulence determinants in CoNS isolates confirmed as etiological factors of human infection. As bacterial virulence and antibiotic resistance have a significant influence on disease severity and treatment, further studies of CoNS virulence, including the species tested in this manuscript, are necessary.

## Figures and Tables

**Figure 1 ijerph-20-05155-f001:**
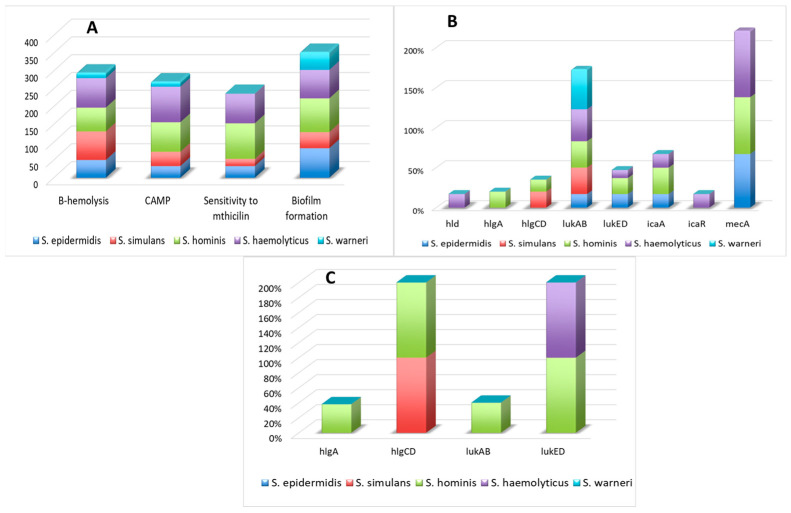
The presence of virulence factors that are homologous to *S. aureus* was tested phenotypically (**A**) and genotypically (**B**). Figure (**C**) presents the expression of detected genes.

**Figure 2 ijerph-20-05155-f002:**
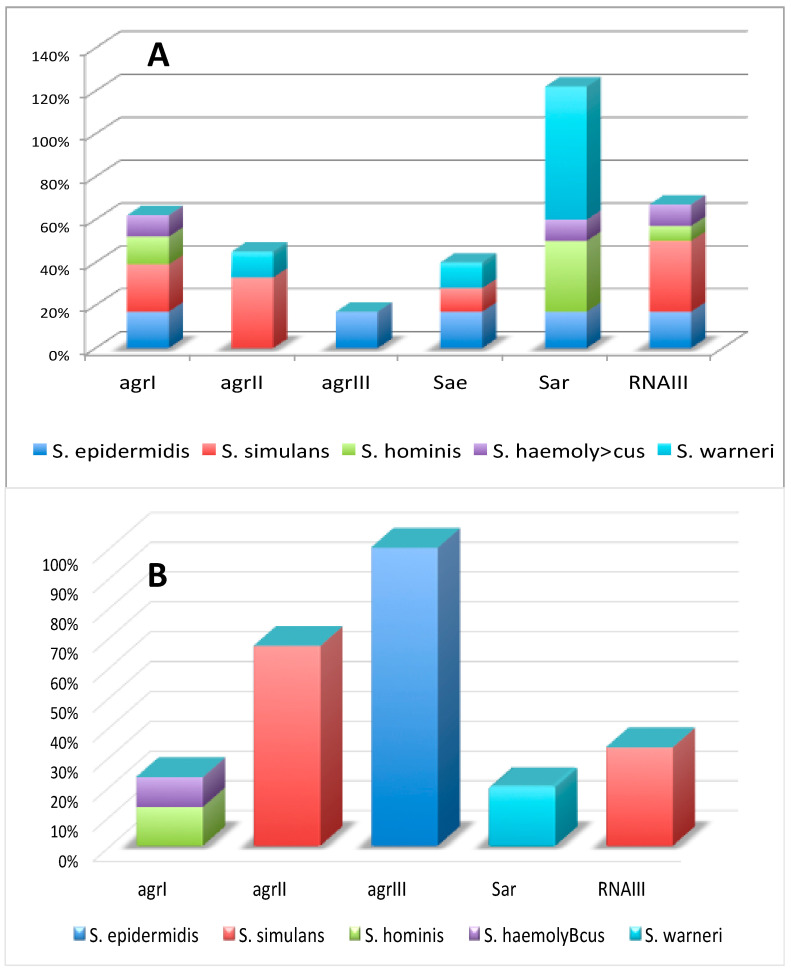
The presence of genes encoding regulators of virulence factors (the accessory gene regulators, (*agr*) (**A**) and their expression (**B**).

**Table 1 ijerph-20-05155-t001:** Primers used for PCR analysis of genes encoding virulence factors.

Lp.	Transcript	Author	Sequence	Size [bp]
1.	*sea*	[25]	5′-GCAGGGAACAGCTTTAGGCGT-3′ 5′-TCTGTAGAAGTATGAAACACG-3′	520
2.	*seb*	[25]	5′-ATGTAATTTTGATATTCGCAGTG-3′ 5′-TGCAGGCATCATATCATACCA-3′	643
3.	*sei*	[25]	5′-CAACTCGAATTTTCAACAGGTAC-3′ 5′-CAGGCAGTCCATCTCCTG-3′	465
4.	*seg*	[25]	5′-CGTCTCCACCTGTTGAAGG-3′ 5′-CCAAGTGATTGTCTATTGTCG-3′	327
5.	*Hla*	[26]	5′-CTTTCCAGCCTACTTTTTTATCAGT-3′ 5′-CTGATTACTATCCAAGAAATTCGATTG-3′	209
6.	*Hlb*	[26]	-GTTGATGAGTAGCTACCTTCAGT-3′ -GTGCACTTACTGACAATAGTGC-3′	309
7.	*hld*	[26]	5′-TTAGTGAATTTGTTCACTGTGTCGA-3′ 5′-AAGAATTTTTATCTTAATTAAGGAAGGAGTG-3′	111
8.	*hlg*	[26]	5′-CACCAAATGTATAGCCTAAAGTA-3′ 5′-GTCAYAGAGTCCATAATGCATTTAA-3′	535
9.	*hlg2*	[26]	5′-ATAGTCATTAGGATTAGGTTTCACAAAG-3′ 5′-GACATAGAGTCCATAATGCATTYGT-3′	390
10.	*hlgCB*	[26]	5′-GCCAATCCGTTATTAGAAAATGC-3′ 5′-CCATAGAYGTAGCAACGGAT-3′	938
11.	*lukED*	[26]	5′-TGAAAAAGGTTCAAAGTTGATACGAG-3′ 5′-TGTATTCGATAGCAAAAGCAGTGCA-3′	269
12.	*lukAB*	[26]	5′-TCACTTCTCCACCATACTTC-3′ 5′-TATCAGCAGCAACGACTC-3′	638
13.	*pvl*	[27]	5′-ATCATTAGGTAAAATGTCTGGACATGATCCA-3′ 5′-GCATCAASTGTATTGGATAGCAAAAGC-3′	433

**Table 2 ijerph-20-05155-t002:** Primers used for PCR analysis of genes responsible for regulation of virulence.

Lp.	Transcript	Author	Sequence	Size [bp]
1.	*sarA*	[30]	5′-TGG TCA CTT ATG CTG ACA GAT T-3′	313
			5′-TTT GCT TCT GTG ATA CGG TTG-3′	
2.	*Sae*	[31]	5′-TGT GGG GTT CAG GAA TTG TT-3′	680
			5′-ATT GAT GAG AAG GAT GCC CA-3′	
3.	*RNAIII*	[32]	5′-ATGATCACAGAGATGTGA-3′	514
			5′-CTGAGTCCTAGGAAACTAACTC-3′	
4.	*agrI*	[33,34]	5′-GTCACAAGTACTATAAGCTGCGAT-3′	439/441
			5′-ATGCACATGGTGCACATGC-3′	
5.	*agrII*	[33,34]	5′-TATTACTAATTGAAAAGTGGCCATAGC-3′	572/575
			5′-ATGCACATGGTGCACATGC-3′	
6.	*agrIII*	[33,34]	5′-GTAATGTAATAGCTTGTATAATAATACCCAG-3′	321/323
			5′-ATGCACATGGTGCACATGC-3′	
7.	*agrIV*	[33,34]	5′-ATGCACATGGTGCACATGC-3′	657/659
			5′-CGATAATGCCGTAATACCCG3′	

**Table 3 ijerph-20-05155-t003:** Primers used for analysis of the expression of genes encoding virulence factors and main genes involved in regulation of virulence in Staphylococcus by RT-PCR.

Lp.	Transcript	Author	Sequence
1.	*16S rRNA*	[35]	5′- TGAGATGTTGGGTTAAGTCCCGCA-3′
			5′-CGGTTTCGCTGCCCTTTGTATTGT-3′
2.	*hlgA*	[36]	5′-AATCGGAGGCAGTGGCTCATTCAA-3′
			5′-GGACCAGTTGGGTCTTGTGCAAAT-3′
3.	*hlgCB*	[36]	5′-TCGGTGGTAATTTCCAATCAGCCC-3′
			5′-CGAATGAATTCGCTTTGACGCCC-3′
4.	*lukED*	[37]	5′-GAAATGGGGCGTTACTCAAA-3′
			5′-GAATGGCCAAATCATTCGTT-3′
5.	*RNAIII*	[38]	5′-TTTATCTTAATTAAGGAAGGAGTGA-3′
			5′-TGAATTTGTTCACTGTGTCG-3′
6.	*lukAB*	[39]	5′-CGT GGA GCG TTA ACT GGA AAT A -3′
			5′-ACA CCT TTA TGT GAC GTA GAT TGA -3′
7.	*agrI*	[40]	5′-CCAGCTATAATTAGTGGTATTAAGTACAGTAAACT-3′
			5′-AGGACGCGCTATCAAACATTTT-3′
			5′-ATAGGAATTTCGACATTATC-3′
8.	*agrII*	[40]	5′-CAATAGTAACAATTTTAGTGACCATGATCA-3′
			5′-GCAGGATCAGTAGTGTATTTTCTTAAAGTT-3′
			5′-TTGCAACAGTAGGTTTGTT-3′
9.	*agrIII*	[40]	5′-CATTATAACAATTTCACACAGCGTGTT-3′
			5′-GCAAGTGCATAAGAAATTGATACATACA-3′
			5′-ATAGTTCTACCAATCTTTTTGG-3′
10.	*hld*	[41]	5′-AAGAATTTTTATCTTAATTAAGGAAGGAGTG-3′
			5′-TTAGTGAATTTGTTCACTGTGTGA-3′
11.	*sarA*	[42]	5′-GTAATGAGCATGATGAAAGAACTGT-3′
			5′-CGTTGTTTGCTTCAGTGATTCG-3′
12.	*sae*	[43]	5′-CAACCATTGCGATTTCTTTACC-3′
			5′-TTAGCTTTAGGTGCTTGTGG-3′

**Table 4 ijerph-20-05155-t004:** The cross-talk between isolates from the same as well as different species.

Supernatant	Strain	CAMP	Β-Haemolysis	Biofilm
Without supernatant	*S. epidermidis* ATCC12228 *	+	+	+
	*S. aureus* ATCC 25923 *	n/a	+	+
	*S. simulans 01 **	-	+	++
	*S. simulans 02 **	+	+	+++
	*S. epidermidis* */**	-	+	+++
	*S. simulans 03 ***	+	+	+++
	*S. hominis 01 **	+	+	++
	*S. haemolyticus **	+	+	++
	*S. hominis 02 **	+	+	++
	*S. simulans 04 ***	+	+	++
*S. simulans 03*	*S. epidermidis* ATCC12228	+	+	+
	*S. simulans 01*	+	+	+
	*S. simulans 02*	+	+	+
	*S. epidermidis*	+	+	+
*S. hominis 01*	*S. epidermidis* ATCC12228	-	+	+
	*S. simulans 01*	-	+	+
	*S. simulans 02*	-	+	+
	*S. epidermidis*	-	+	+
*S. haemolyticus*	*S. epidermidis* ATCC12228	+	+	+
	*S. simulans 01*	+	+	+
	*S. simulans 02*	+	+	+
	*S. epidermidis*	+	+	+
*S. hominis 02*	*S. epidermidis* ATCC12228	-	+	+
	*S. simulans 01*	-	+	+
	*S. simulans 02*	-	+	+
	*S. epidermidis*	-	+	+
*S. simulans 04*	*S. epidermidis* ATCC12228	+	+	+
	*S. simulans 01*	+	+	+
	*S. simulans 02*	+	+	+
	*S. epidermidis*	+	+	+
*S. epidermidis*	*S. epidermidis* ATCC12228	+	+	+
	*S. simulans 01*	+	+	+
	*S. simulans 02*	+	+	+
*S. aureus* ATCC 25923	*S. epidermidis* ATCC12228	n/a	+	+
	*S. simulans 01*	n/a	+	+
	*S. simulans 02*	n/a	+	+
	*S. epidermidis*	n/a	+	+

* Inactive *agrI* Gene, ** Active *agrIII* or *agrII* Gene. - No Biofilm Producer, + Weak Biofilm Producer, ++ Moderate Biofilm Producer, +++ Strong Biofilm Producer.

## Data Availability

Not applicable.

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
