# Peer review of "Analysis of the Presence of the Virulence and Regulation Genes from Staphylococcus aureus (S. aureus) in Coagulase Negative Staphylococci and the Influence of the Staphylococcal Cross-Talk on Their Functions"

_ijerph, 2023, doi:10.3390/ijerph20065155_

Round 1

Reviewer 1 Report

Dear Authors

Thanks for your manuscript with title: Analysis of the Presence of the Virulence and Regulation Genes from S. Aureus in Coagulase Negative Staphylococci and the In-Fluence of the Staphylococcal Cross-Talk on Their Functions, the review of the aforementioned manuscript has been finished and there are some points about it that theyere most be addressed and you could find at the attached file. 

Best Regards

Author Response

Dear Reviewer, we are very grateful for detailed analysis of our manuscript. All suggestions were taken under consideration and are included in the  manuscript attached. Additionally, response to all suggestions (beside editorial) was noted in the comments section.

Sincerely yours,

Dr. Magdalena Grazul.

Reviewer 2 Report

Dear Authors,

topic was fine presented and it is of sure actuality and interest. I reported some suggestions that I hope could improve the manuscript.

Best

Author Response

(The authors gave the same response as above.)

Round 2

Reviewer 1 Report

Dear Authors

Thanks for revised version. The review of the revised version has been finished and most of the comments has been addressed. Best Regrads